# International Health Cooperation in the Post-Pandemic Era: Possibilities for and Limitations of Middle Powers in International Cooperation

Yongmin Kim [1] and Youngdeuk Park [2,*]

1    Konkuk University China Institute (KUCI), Konkuk University, Seoul 04020, Korea; kym7224@konkuk.ac.kr
2    Department of Political Science and Diplomacy, Chungnam National University, Daejeon 34134, Korea
*    Correspondence: y_park@cnu.ac.kr

**Abstract:** The COVID-19 pandemic has left international cooperation and liberalistic values in crisis. As liberalism's downfall is widely discussed, international collaborations like the European Union are criticised for their inability to operate adequately during the pandemic. The four examples in this paper are middle power countries (South Korea, Japan, Taiwan, and Singapore) in terms of economic scale and influence. The purpose of this study was to uncover possibilities for and limitations of these middle powers within international cooperative efforts during and after the pandemic. The unknown factor is the path the post-pandemic world will follow. Will nations focus on independent survival? Or will international cooperation shape the new world? Globalisation already seems to have progressed too far for the national egoism of the great powers to prevail. Even if face-to-face is replaced by virtual and offline meetings move online, the social nature of humans remains unchanged, and international cooperation remains valid. The four middle power countries in Asia, which are included among the most economically successful countries, are important to international society based on their relatively excellent quarantine performance. What is important in the diplomacy of middle power countries is not traditional security and hard power but the soft power of international law, human rights, health security, and international cooperation.

**Keywords:** COVID-19; health security; middle power; international cooperation; post-pandemic



## 1. Introduction

As a result of the COVID-19 global pandemic, international cooperation and the values represented by liberalism have fallen into crisis. As the downfall of liberalism is being widely discussed, organisations representing international collaborations, such as the European Union, are being criticised for their inability to operate adequately during the pandemic crisis. In addition to the crisis, the largely economic tension between two superpowers, the United States and China, has adversely affected areas that urgently need support through international cooperation, including the development and distribution of a vaccine. However, few studies have examined the responsibilities incumbent upon the so-called middle power countries in East Asia, represented by Korea and Japan in this situation. Therefore, in this paper, we examine the strengths and limitations of middle power countries in a pandemic situation through a comparative analysis that centres on the cases of Korea and Japan, which are representative middle-sized countries, and incorporates the cases of Taiwan and Singapore. Taiwan and Singapore are small countries by their population, but they have enough economic power as middle power countries. Some may consider such a discussion premature because the coronavirus is not yet under control, but the present time, when the policy on COVID-19 is changing, is, in fact, the perfect time to discuss post-pandemic international cooperation in order to be adequately prepared. The four countries selected as examples in this paper are countries that are difficult to choose unilaterally for various reasons, such as geography and economy, in the context

of the United States-China confrontation; moreover, they are middle power countries in terms of economic scale and influence. Therefore, by comparing and analysing countries that are similar in these ways and that have relatively superior coronavirus quarantine performance compared to Europe and the United States, we endeavour to predict the direction of middle power diplomacy in the international community after the COVID-19 pandemic has subsided and consider the possibilities for and limitations of that diplomacy.

## 2. Literature Review

Global cooperation on health-related matters has a long history, as exemplified by more than 70 years of activity from the World Health Organization (WHO). However, the topic of health security that has emerged has a relatively short history, starting with the United Nations Development Programme's (UNDP's) human security discussion in 1994. As health security is recognised as a basic human right[1], attention has been focused recently on international health cooperation in developing countries and conflict areas where such rights are not guaranteed. Efforts to overcome global inequalities caused by the COVID-19 pandemic illustrate global health cooperation.

Disease-causing bacteria and viruses are spread through human contact, whether through war, travel, or trade. As long as the world remains interconnected, infectious diseases will remain fundamentally international issues. The Black Death pandemic reminded humanity of the importance of the quarantine, and in the sixteenth century, some governments built health certificate systems (Zacher and Keefe 2008). Cholera, a native Indian epidemic, spread to Europe via merchants, British troops stationed in India, and pilgrims passing through India. At the time, European governments attempted to combat cholera by improving hygiene, as they were unaware of the specific cause of and cure for the disease. However, each country has its own requirements for quarantine and hygiene. Eventually, certain Western European countries recognised the importance of international cooperation in properly responding to the epidemic (Howard-Jones 1975).

The International Sanitary Conference (ISC) constituted the first international initiative intended to enhance public health in 1851. However, the ISC was unable to produce substantial results due to a lack of coordination among national interests concerned about potential economic losses caused by quarantine measures and the underdevelopment of medical research in the nineteenth century (Baldwin 2005; Birn 2009; Howard-Jones 1975). Nevertheless, as medical science advanced in the early twentieth century, countermeasures against infectious diseases were developed.

As worldwide trade increased, so did the importance of international efforts to combat infectious diseases. Eventually, in 1948, the WHO was founded as an international organisation devoted to global health issues. The WHO established the International Sanitary Regulations, which were reorganised in 1969 as the International Health Regulations (IHR). Although the HIS was revised in the wake of the Severe Acute Respiratory Syndrome (SARS) outbreak, its lack of success in addressing the COVID-19 pandemic reflects the limitations of institutional cooperation.

The global health cooperation system did not sufficiently contribute to resolving the COVID-19 crisis because the conflicts and interests related to the powers applied in international relations were reflected in the international governance structure, which led to a decline in trust in international organisations such as the WHO. Alternatives should be sought, as similar pandemics or various crises may occur in the future. Thus, in this study, we set out to clarify the role the middle power countries should play in providing these alternatives and to assess the appropriateness of these suggestions.

Previous studies that investigated the possibilities for and limitations of middle powers in the context of international cooperation can be divided into two broad categories. The first group of studies focuses on the diplomacy of middle power countries within the traditional concept of international cooperation and liberalism, and the second includes those that discuss the post-pandemic international order. In a recent study on international cooperation concerning middle power diplomacy and the liberalism international relations

theory, Professor Soeya Yoshihide of Japan applied the concept of middle power diplomacy when he discussed Japanese diplomacy. This theory was considered in the context of Japan, where the peace constitution prohibits the forming of an army and the concept of human security, not traditional security, which is emphasised as a national strength. Yoshihide also asserted that middle power countries in the East Asian region must cooperate with other middle power countries through linkages to survive confrontations with the great powers, such as during the United States-Soviet Cold War (Soeya 2013).

There were several theories discussed in international relations. We have to mention the concepts of soft power and international cooperation against political realism and hard powers. The main four countries we discussed in this article (Japan, South Korea, Singapore, and Taiwan) are, to a large extent, models of state organisation of soft power. Such concepts are well established in theory by authors and works, such as several works by Joseph Nye (2004, 2008, 2011, 2015). However, political realism researchers, represented by Morgenthau (2006) and Waltz (2008), have also criticised international cooperation and liberalism.

Recently, attention has returned to this subject due to the intensification of the United States-China confrontation. In discussions on the crisis of liberalism and international order since Trump was elected, many scholars have expressed concern about Trump's 'America First!' policy. Ikenberry, a leading theorist on liberalism, cited the United States' loss of its ultra-superpower status, China's non-liberalistic international politics, and the emergence of the North Korean nuclear threat as causes of the crisis (Drezner 2019; Funabashi and Ikenberry 2020; Norrlof 2018). The United States is expected to change its position of isolation following Biden's victory and return to its traditional diplomatic stance that emphasises alliances. However, such a transition still leaves the failure of liberalism due to the collapse of the global product chain triggered by the United States-China confrontation and the COVID-19 pandemic, as well as the inability of international organisations to operate during the pandemic crisis, unaddressed. Thus, liberalism and international cooperation are likely to remain in crisis. Various topics are being discussed in studies on the possibilities for and limitations of middle powers in facilitating international cooperation concerning the international order in the post-pandemic period. Multiple broader studies have examined whether we can truly return to the world as it existed before COVID-19. This ongoing discussion is limited neither to international politics nor to diplomacy forums; indeed, various discourses are unfolding related to all aspects of the post-pandemic world that lies ahead. However, in this paper, only prior studies from the international cooperation and diplomacy sectors are summarised.

A fiercely debated topic related to the post-pandemic international order on the fundamental discourse level (Kiran 2020) is whether the world after COVID-19 will be a world of cooperation or a world of competition. One study predicted various problems in global governance after the pandemic crisis (Chaudhry et al. 2020), and another considered problems that will occur in the process of recovery from the economic crisis that has accompanied the pandemic (Song and Zhou 2020). A study on the possibility of change in the international order triggered by the pandemic and the United States (Nye 2020) and a study on international relations after the COVID-19 pandemic ends have also made worthwhile contributions to the literature. In addition, some studies (e.g., Brown and Susskind 2020) have sought to identify best practices for international cooperation during the coronavirus outbreak, while others (e.g., Anderson et al. 2021) have discussed the so-called 'new normal' that will settle in after most traces of the pandemic have vanished.

In conclusion, the numerous opinions about the opacity of the post-pandemic international order diverge in various directions; similarly, opinions are divided as to whether the pandemic will trigger a paradigm shift that shakes the international political order to any significant degree, or whether it will ultimately effect only a small change. Conflicting opinions are erupting because the approach to the issue of international order naturally differs depending on the situation, region, and national power of the country or countries involved. Therefore, in this paper, we discuss the strengths and weaknesses of four middle-sized countries in Asia in relation to international cooperation in the post-pandemic era,

reflecting on the discussions of previous studies. Next, in Section 3, we discuss in detail the strengths of the four middle power Asian countries relative to international cooperation during the COVID-19 pandemic.

### 3. Strengths of Middle Power Asian Countries Related to International Cooperation and the COVID-19 Pandemic

What advantages do the middle power countries in Asia have in a world shaped by the COVID-19 pandemic? The first strength is the excellence of their coronavirus prevention efforts. Although Asian middle power countries are experiencing an increase in infections during the third wave of the virus as of the writing of this paper, with spring 2022 approaching, they are demonstrating comparatively superior quarantine performance compared to that of the United States and advanced European countries. Figure 1 presents statistics on COVID-19 infections in major countries as of 4 February 2022. The differences in the daily numbers of new infections in the representative developed countries—the United States, the United Kingdom, Germany, and France—and in the countries representing the middle powers—Korea, Japan, Taiwan, and Singapore—are significant, as clearly illustrated in Figure 1. Additionally, the four Asian countries have been more successful at preventing COVID-19 fatalities. Figure 2 depicts the excess mortality rate, which serves as a measure for the number of casualties caused by COVID-19. The data indicate which countries have been more successful in preserving lives. Various factors, such as mask wearing, lifestyles and culture, as well as differences in medical systems, may impact these numbers, but objectively, the outstanding quarantine performance of the four countries representing Asia's middle powers cannot be disputed.

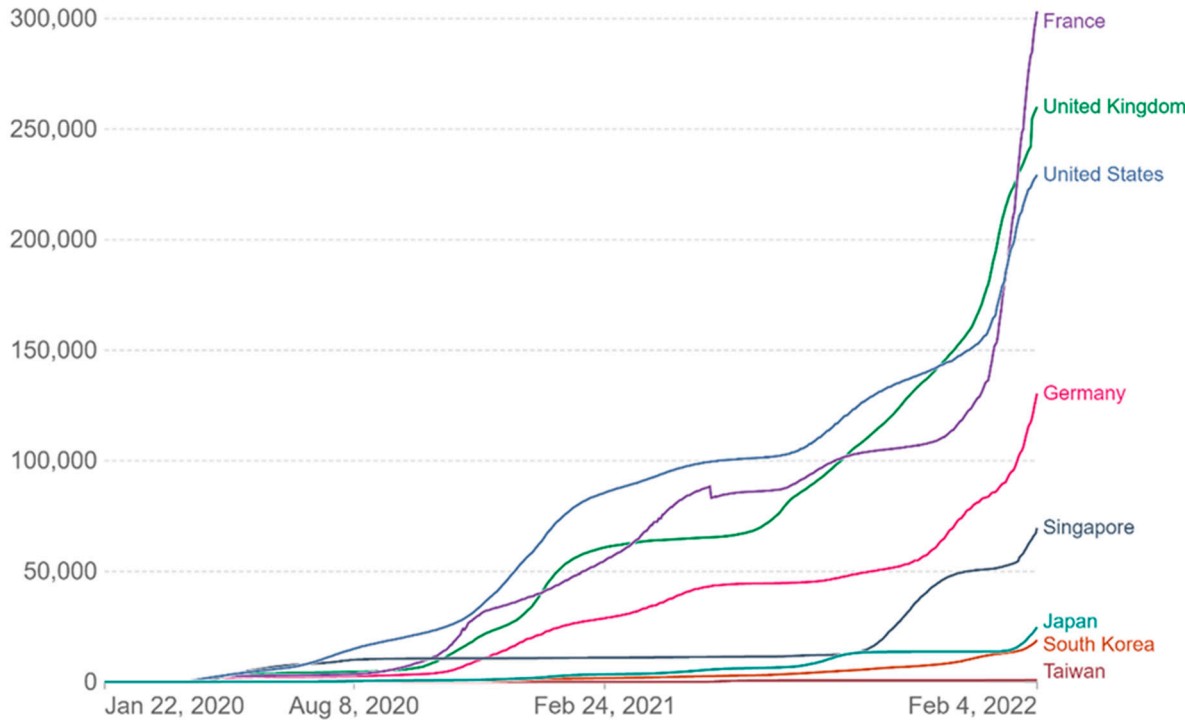

**Figure 1.** Daily new confirmed COVID-19 cases in developed and middle power countries (per million people). Source: Data from Johns Hopkins University CSSE COVID-19 Data via Our World in Data (https://ourworldindata.org (accessed on 9 January 2022))[2].

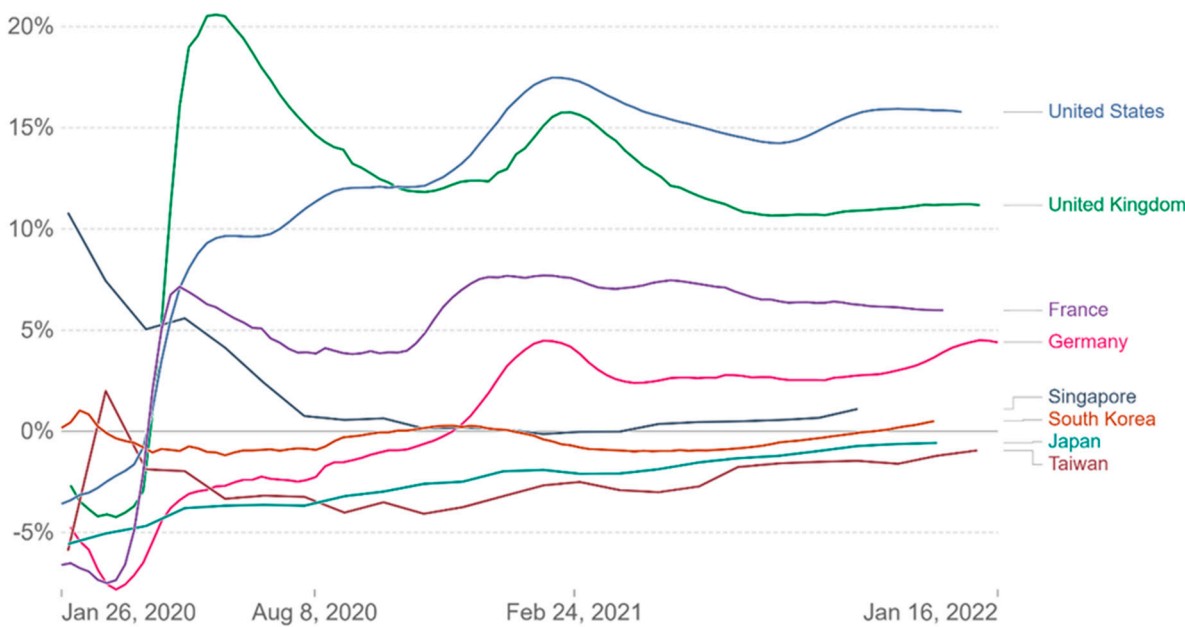

**Figure 2.** Excess mortality rates in developed and middle power countries. Source: Data from Human Mortality Database (2022), World Mortality Dataset (2022) via Our World in Data (https://ourworldindata.org (accessed on 9 January 2022)) (Karlinsky and Kobak 2021).

The success of national quarantine efforts immediately led to a relative reduction in economic damage. Figure 3 illustrates this. Except for the United States, which has suffered relatively little economic damage despite the large number of COVID-19 deaths in that country, the economic growth rates of the Asian middle power countries that are controlling the number of deaths from the coronavirus are showing comparatively minimal declines. As Figure 2 demonstrates, Britain, Italy, and France recorded severe negative growth along with many deaths. These statistics reflect a characteristic of the COVID-19 pandemic, during which quarantines not only impact the spread of infectious diseases but also affect the national economy. If a country's quarantine policies are successful in minimising economic damage, that country can expect to recover after the pandemic has ended much more quickly than other countries. Resilience will be an important post-pandemic theme as, in addition to preventing the spread of the disease and maintaining the medical systems, the quarantines instituted also played a role in the broader war over comprehensive national power. The success of these four Asian middle power countries is demonstrated by their relative economic stability and technological developments. In fact, Korea and Japan have the ability to mass-produce and distribute sufficient COVID-19 vaccines worldwide and can compensate for the weaknesses of the COVAX facility[3].

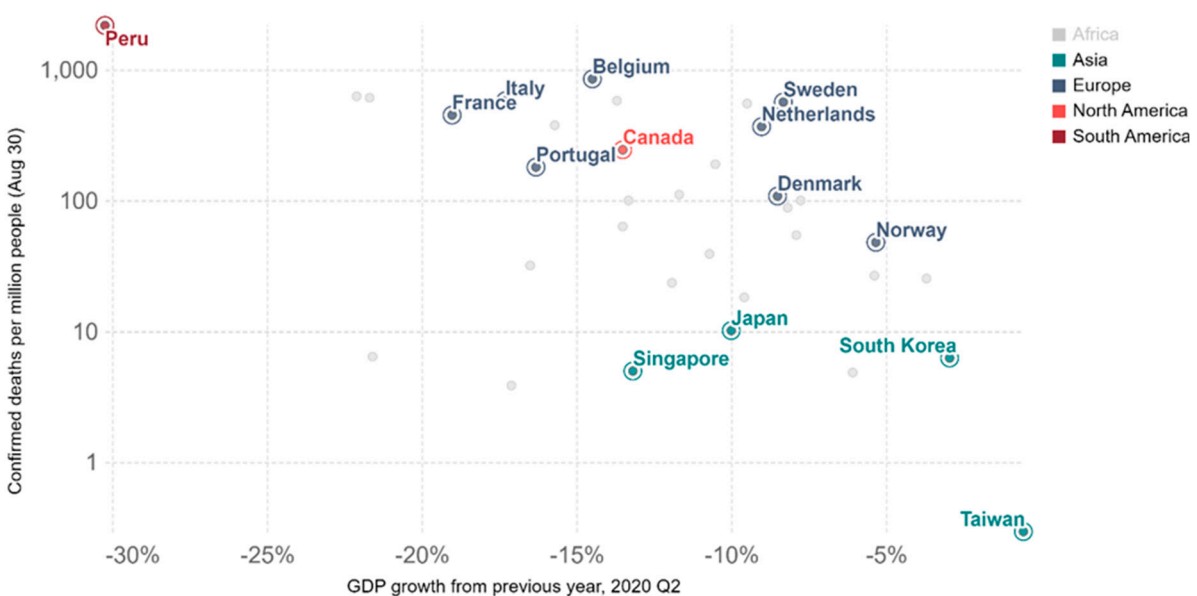

**Figure 3.** Economic decline in 2020 s quarter vs. confirmed deaths due to COVID-19. Source: Data from Johns Hopkins University CSSE via Our World in Data (https://ourworldindata.org (accessed on 9 January 2022)).

The remarkable economic and technological competitiveness of the four Asian middle power countries represents the second advantage they have in reviving international health cooperation. Figure 3 and Table 1 summarise the economic and technological strength of the four countries. As mentioned previously, the negative economic impact experienced by these countries, compared to what other countries have experienced, has been relatively small due to their successful quarantine measures, so their resilience will be strong, which gives them an advantage because they will be able to use their existing systems as is. As shown in Table 1, the four Asian middle power countries are leaders in patent applications and technological innovations. Additionally, these countries are significant players in the global economy, accounting for a sizable portion of international trade.

**Table 1.** Technological, Economic Advantages of Asian Middle Power Countries.

|  | Global Innovation Index Score (2021) | Total Patent Applications by Origin (2020) | Merchandise Exports (2020, Million USD) |
|---|---|---|---|
| Rep. of Korea | 59.3 (5) | 260,614 (4) | 512,498 (7) |
| Japan | 54.5 (13) | 423,264 (3) | 641,318 (5) |
| Singapore | 57.8 (8) | 7946 (25) | 362,534 (14) |
| Taiwan | - | - | 347,192 (15) |
| USA | 61.3 (3) | 496,123 (2) | 1,424,935 (2) |
| Germany | 57.3 (10) | 168,092 (5) | 1,382,532 (3) |
| China | 54.8 (12) | 1,441,086 (1) | 2,589,952 (1) |
| France | 55.0 (11) | 64,287 (6) | 488,637 (9) |
| UK | 59.8 (4) | 53,079 (7) | 399,530 (12) |

Source: World Intellectual Property Organization (2021).

Some expect that lifestyle changes prompted by the COVID-19 pandemic will reshape our society into a completely cyber and non-contact existence. However, international cooperation on existing production methods cannot completely disappear. Indeed, when vaccines and treatments become available, these production methods are intended to

sequentially distribute the medicines to the world. Hence, repair of the supply chain is essential. The middle powers should lead efforts to connect superpowers, such as the United States and China, with developing countries during difficult situations.

For example, middle power countries should link with other countries regarding the COVAX Facility (except Japan, which has already secured enough vaccines through pre-purchase). As it currently stands, nine Coalition for Epidemic Preparedness Innovations (CEPI)-supported candidate vaccines are part of the COVAX initiative, with an additional nine candidates under evaluation. Moreover, procurement conversations are ongoing with additional producers who are not currently receiving research and development funding through COVAX. This gives COVAX the largest and most diverse COVID-19 vaccine portfolio in the world, as 80 potentially self-financing countries have submitted non-binding expressions of interest to the Gavi-coordinated COVAX Facility, joining 92 low- and middle-income economies that are eligible to be supported by the COVAX Advance Market Commitment (AMC). The goal of bringing the pandemic under control via equitable access to COVID-19 vaccines needs urgent, broadscale commitment and investment from countries (WHO 2020). This commitment and investment will be integral to post-pandemic international cooperation as a measure to restore trust in international organisations that have been criticised for their response to the COVID-19 pandemic. The best countries to serve as the link between the 92 developing and advanced economies are the Asian middle powers. COVAX set a goal to deliver, by the end of 2021, two billion doses of safe, effective vaccines that had passed regulatory approval and/or WHO prequalification to be offered equally to all participating countries, proportional to their populations, initially prioritising healthcare workers and then expanding to cover vulnerable groups, such as the elderly and those with pre-existing conditions. The plan was to make further doses available based on a country's need, vulnerability, and COVID-19 threat. The COVAX Facility would also maintain a buffer of doses for emergency and humanitarian use, including dealing with severe outbreaks before they spiral out of control (WHO 2020). If this goal is met, expectations for multilateral governance and international organisations in international cooperation can be restored. Vaccines and therapeutics are the most urgent matters, but the same logic applies to other economic productions.

Another strength of the four countries representing Asian middle power nations relates to their being free countries that are relatively unaccountable for the current lack of international leadership, the lack of United States-China cooperation, and the inefficiency of international organisations in dealing with the COVID-19 pandemic. They also maintain close relations with the United States and China, so they are countries with plenty of space to move. All four countries have close economic relationships with China, and on the matter of security, Korea and Japan have close relations with the United States. COVID-19 issues require so-called niche diplomacy (Maitre 2018), which is easier for middle powers to manoeuvre within, compared to diplomacy related to traditional security issues, such as military security, in which the sharp interests of the powers collide. As noted previously, the ways in which the four countries in focus in this paper responded to COVID-19 have attracted global attention as examples of successful cases for containing the coronavirus, which gives them an advantage in developing international cooperation diplomacy. Non-traditional security, especially human security, has become a major security agenda item in the international community (Gilder 2021), where a strong trend towards establishing post-pandemic international cooperation has remained secondary politically to the position of traditional military security prior to COVID-19. In such a situation, a hard power country with strong military and economic power will have difficulty taking the lead in everything; therefore, the role of countries with crisis response power becomes more important. In addition, as the Trump administration has been criticised for its response to COVID-19, the United States has not been the centre of rapid response, and the international community's distrust of China's transparency is still high; thus, expectations for middle power countries are bound to grow (Lee and Piper 2020).

The most important thing the world has learned from COVID-19 is that border blocking is not the best solution, evidenced by the continued spread of infections within closed borders. As such, countermeasures should be enacted by promptly sharing infectious disease information and quarantine know-how with each country; manufacturing quarantine equipment, facilities, and drugs in large quantities; and immediately supplying them where necessary. Only then can the function of the social community be restored quickly without being paralysed, preventing further confusion and fear. However, from the perspective of the zero-sum game of traditional military security, cooperation is difficult due to low trust towards other countries. Therefore, middle power Asian countries, not the superpowers participating in the zero-sum game, must actively lead the discussion.

Not only are Asian middle power countries effective, but they have also been normative in their response to the pandemic. The Pandemic Violation of Democratic Standards Index Score is a measure of how much a country's response to the pandemic eroded democratic principles. The level of pandemic backsliding measured by V-Dem is shown in Table 2. According to the data, Asian middle power countries, with the exception of Singapore, an authoritarian country, responded well to COVID-19 with few to no violations of democratic values. On this basis, the assertion can be made that the middle power countries benefit from exerting leadership in international health cooperation.

**Table 2.** Pandemic Violations of Democratic Standards Index Score.

| | Pandemic Violations of Democratic Standards Index Score | Level |
|---|---|---|
| Rep. of Korea | 0 (142/144) | No Violations |
| Japan | 0.1 (125/144) | Minor Violations |
| Singapore | 0.3 (59/144) | Moderate Violations |
| Taiwan | 0 (144/144) | No Violations |
| USA | 0.2 (101/144) | Moderate Violations |
| Germany | 0 (138/144) | No Violations |
| China | 0.75 (1/144) | Major Violations |
| France | 0.1 (123/144) | Minor Violations |
| UK | 0.15 (118/144) | Moderate Violations |

Source: V-Dem Institute, 2022. Pandemic Backsliding Project.

Korea and Japan, in particular, have been actively assuming the roles that affluent countries are expected to play in the international community. Korea became an independent country following World War II; at the time, it was the world's poorest country, but now, Korea is a country that provides official development assistance (ODA). Korea's development assistance will total $2.25 billion in 2020 in ODA grant-equivalent terms. Japan provides about $16.26 billion in aid based on ODA grant equivalents, ranking fourth among DAC countries (OECD 2022).

To summarise, the three main strengths of Asian middle power countries are that they achieved excellent results from quarantine efforts; they possess economic power, science and technology, and manpower with the ability to restore the global supply chain. They can pursue international cooperation with a non-traditional security and human security approach rather than with the superpowers' approach of extreme confrontation, relatively free from the responsibility of zero-sum competition. In this way, mobility and trust can be promoted through these middle power countries.

Next, in Section 4, we discuss the limitations of the middle powers in Asia.

## 4. Limitations of Middle Powers in International Cooperation in COVID-19

Having examined the strengths of four middle power Asian countries related to international cooperation in the post-pandemic era, the limitations of these nations—Korea,

Japan, Taiwan, and Singapore—must also be considered. The first limitation is that they are never free from the United States-China confrontation. Each of the four countries has close economic relations with China and is highly dependent on exports, while Korea and Japan have long-standing alliances with the United States in terms of security. While the goal is to balance and talk about the separation route of the United States for security and China for the economy, if the United States-China confrontation intensifies and forces other countries to choose between the two great powers, all four middle powers will face difficulties. For example, Korea and Japan militarily maintain strong cooperation with the United States, but at the same time, they participate in the China-led Regional Comprehensive Economic Partnership (RCEP) (Crivelli and Inama 2022) for economic reasons. RCEP is a free trade agreement between the Asia-Pacific nations of Australia, Brunei, Cambodia, China, Indonesia, Japan, Laos, Malaysia, Myanmar, New Zealand, the Philippines, Singapore, South Korea, Thailand, and Vietnam. As of 2020, the 15 member countries accounted for about 30% of the world's population, with 2.2 billion people and 30% of the global GDP at $26.2 trillion, making it the biggest trade bloc in history. The RCEP is the first free trade agreement between China, Japan, and South Korea, three of the four largest economies in Asia. At the time it was signed, analysts predicted that it would help stimulate the economy amid the COVID-19 pandemic, as well as 'pull the economic centre of gravity back towards Asia' and amplify the decline of the United States in economic and political affairs (Gunia 2020). Even Japan, which was not enthusiastic about signing prior to doing so, cannot ignore the fact that members of the RCEP account for around a third of the world's population and global GDP (Kim 2021). This new free trade block will be bigger than both the area covered under the United States–Mexico–Canada Agreement and the European Union. According to the future forecast, continued economic growth, particularly in China and Indonesia, may lead the total GDP in the original RCEP membership to grow to over $100 trillion by 2050, roughly double the project size of the Trans-Pacific Partnership (TPP) economies. Japan's options related to joining the RCEP may have disappeared when Trump withdrew the United States from the TPP agreement. Such a case is also possible if the United States-China confrontation intensifies, narrowing the choices of the middle powers along the way. When traditional security issues emerge, even more difficulties will arise, and in Japan, potential territorial disputes and even deterrence may become issues of concern. Therefore, even if the confrontation between the United States and China deepens, it is important to prepare a space for middle power countries to respond to it. The strengthening of relations with India and Australia, currently being promoted by Japan, serve as a good example for the middle power countries in the region.

The second limitation of the middle power Asian countries concerns whether the successful advancement of international cooperation can be hindered by a deterioration in the mutual relations between the middle power countries. This is well represented by the relationship between Korea and Japan. Currently, the relationship between these two nations has weakened to such an extent that it has been deemed the worst in history since the normalisation of diplomatic relations in 1965. If such bilateral relations continue to worsen, promoting international cooperation will be difficult. Diplomatic relations between Korea and Taiwan were cut off after the rise of China, limiting communication between the two nations to only private exchanges, and if China applies pressure, cooperation will not be smooth. Since Korea, Japan, Taiwan, and Singapore each have their own domestic issues that often overlap economically, conflicts of interest that can lead to various issues between these countries exist. As part of Japanese diplomacy, including within Korea-Japan relations, Japan has long aimed to establish Japanese-style human security (non-traditional security).

The concept of security as it is traditionally defined is diminished in Japanese-style human security. Japan introduced Japanese-style human security in the 1994 UNDP Human Development Report, emphasising the concepts of both 'freedom from fear' and 'freedom from want'. The report identifies four basic characteristics (global, anthropocentric, interdependent and early preventive) and seven central elements (economy, food, health,

environment, individual, community and political). Since the UN first advocated for human security in 1994, Japan has taken on an international role in actively promoting human security through economic support for the third world, domestically and internationally. Indeed, human security has been promoted as a major factor in Japanese diplomacy.

While claims have been made that the proportion of human security in Japanese diplomacy has decreased since the 2000s, its importance was re-examined following the 2011 East Japan earthquake, and it has reached the present level, among the soft issues of human security (Hynek 2012). In that sense, the possibility of international cooperation related to non-traditional security can be explored, but when a traditional security issue arises, it has been given lower priority. That the COVID-19 pandemic represents a typical health security issue is helpful because it increases the importance of human security. Therefore, improving relations between middle power countries is essential to demonstrate their strength in diplomacy with other middle power countries after COVID-19 is contained.

As such, a plan to improve the Korea-Japan relationship has been developed. In the context of the COVID-19 pandemic, the trend in Korea has been to underestimate the necessity of cooperation with Japan. As Japan did not demonstrate its prestige as an economic powerhouse in the previous Cold War or in the early 1990s, the country is not as important to the Republic of Korea's diplomacy as it previously was. Nevertheless, recent negotiations with North Korea—the series of so-called 'Japan passing' controversies, such as the Republic of Korea and inter-Korean dialogues led by the United States—and the series of conflicting phases after the Supreme Court decision are not desirable for either country in the long term. To realise the two countries' common goal of denuclearising North Korea, the South Korean government must provide a space for Japan to play a constructive role. In the short term, humanitarian economic support is needed to resolve North Korea-Japan issues, including the kidnapping problem. For a long-term resolution, after diplomatic relations in Japan are normalised, it can consider providing incentives to North Korea for large-scale economic aid. As a similar example, in the past of Koizumi's cabinet, the Kim Dae-Jung administration needed to refer to the role of the 'intermediary' between Korea and Japan. Although North Korea does not officially report confirmed coronavirus cases, the country is known to be in a difficult situation due to the COVID-19 pandemic and economic sanctions (BBC 2022). Considering this, if South Korea cooperated closely with Japan, where abductee issues are pending, then Japan can be attracted as a supporter of the Korean Peninsula Initiative of Peace and Prosperity (Unikorea 2022), which the Korean government put forward with the aim of creating a new Korean Peninsula regime. This would ultimately benefit Japan as well, as Japan is also hoping for a stable Korean Peninsula. Therefore, if the relationship is rebuilt through low-level cooperation centred on various soft issues in place of the current state of confrontation between Korea and Japan, and the development of the relationship progresses to encompass hard issues, including the denuclearisation of North Korea in the long term, cooperation in the post-pandemic context will lead to regional stability to the degree that can be expected. As such, finding issues that each middle power country can cooperate on is vital; this applies to all middle power countries in the region. If cooperation is promoted in myriad ways related to these issues, damage can be minimised, even if a conflict occurs due to the individual interests of a participating country. Apart from such a conflict between Korea and Japan, hatred against China in Korea and Japan has risen throughout the COVID-19 pandemic, as shown in Figure 4, and is at a record high in other developed countries. This is a potential risk factor considering East Asia's regional characteristics.

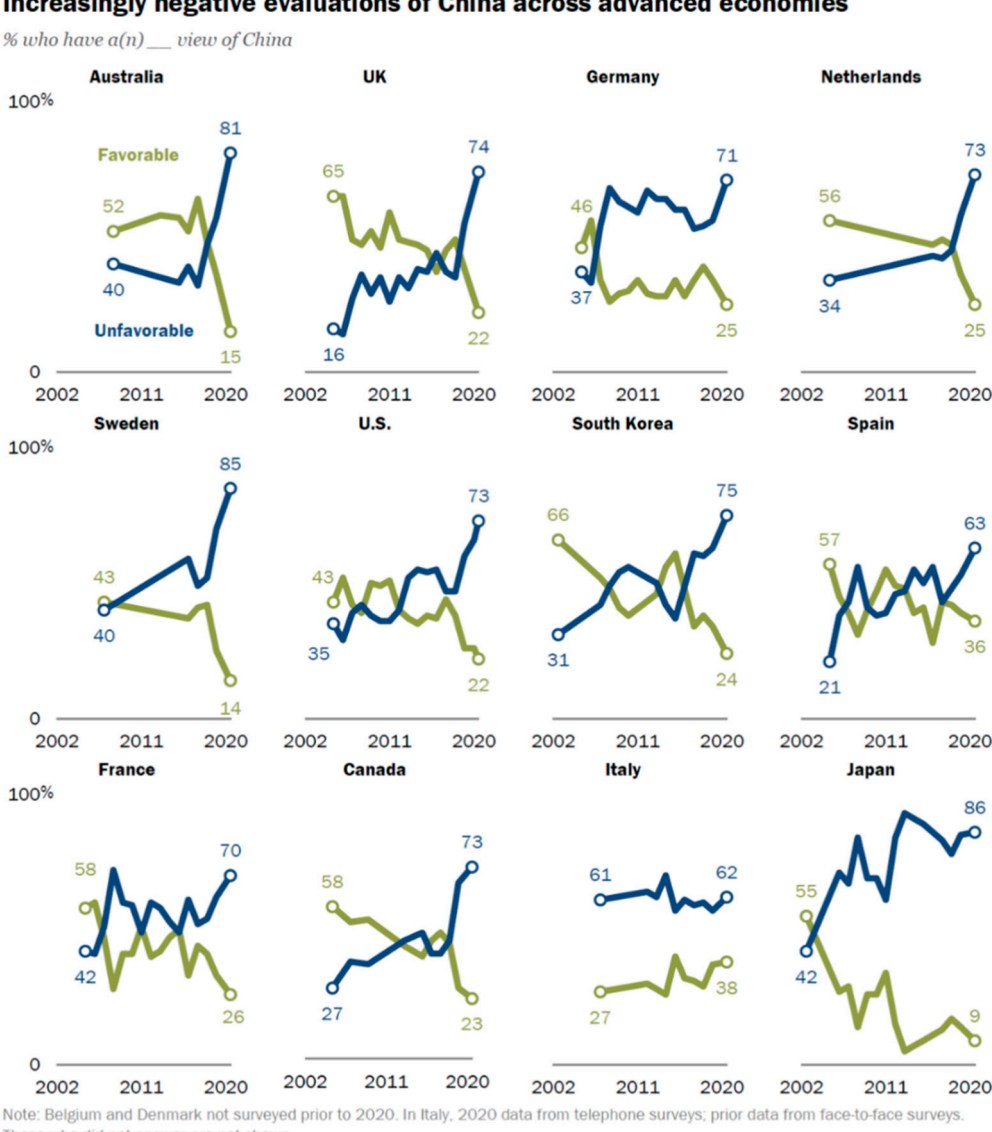

**Figure 4.** Increasingly negative evaluations of China across advanced economies. Source: Pew Research Center.

The third limitation of the middle power countries related to fostering international cooperation in the post-pandemic period is the deterioration of international organisations that have been criticised for failing to function properly during the COVID-19 crisis. The role of the WHO, which has been criticised for its handling of the pandemic, should be re-evaluated, as it is essential for international organisations to exert diplomatic influence differently from the United States and China. The U.S. government stopped funding the WHO in April 2020, damaging the multilateral platform for global health diplomacy and its secretariat's programmes to promote international health cooperation. Then, the United States announced its decision to withdraw its membership from the WHO in May 2020 and gave formal notice in July 2020 of plans to withdraw in July 2021, challenging the legitimacy of the UN specialised agency. However, this situation is expected to change under the Biden administration when he officially becomes America's president. Still, many people agree that the WHO needs to go through a series of reforms (Lee and Piper 2020). After the outbreak of the Ebola virus disease in West Africa in 2013-2016, an intensive discussion took

place on efforts to reform the WHO secretariat, to create emergency funds within the WHO and the World Bank, and to coordinate between the WHO's health emergency response and the UN's humanitarian operations. This time, beginning in August 2020, France and Germany took the lead in initiating another round of discussions on WHO reform to strengthen the UN specialised agency's legal power and financial resources. As a short-term solution, the WHO must become more effective in harmonising and coordinating multiple initiatives taken by its member states and non-state actors to combat COVID-19. East Asian countries can strengthen multilateralism in the fight against COVID-19 by supporting the WHO and its regional offices for the Western Pacific and Southeast Asia, both politically and financially. Non-state actors in East Asia can also support the WHO's response to COVID-19. The COVID-19 Solidarity Response Fund for the WHO was created by the UN Foundation and the Swiss Philanthropy Foundation in March 2020 to collect donations from individuals and organisations in support of the WHO and its partners, such as the UN Children's Fund (UNICEF), the World Food Programme (WFP), the UN High Commissioner for Refugees (UNHCR), the United Nations Relief and Works Agency for Palestine Refugees in the Near East (UNRWA) and the Coalition for Epidemic Preparedness Innovations (CEPI). More than 651,000 individuals, companies, and philanthropies contributed or pledged $238 million to the COVID-19 Solidarity Response Fund for WHO as of November 2020.

For the same reason, the World Trade Organization's (WTO's) role is also particularly important. The middle power countries need to actively communicate with the superpower countries about the worldwide dissemination of vaccines and treatments from a human rights perspective. The need to revive the economy and escape from the COVID-19 crisis is urgent. Falling into self-interest without thinking about the values of middle power nations, such as human rights, international law, and international cooperation, may help overcome the crisis in the short term, but in the long term, such an approach will not be able to overcome the impact on the international community and will not be able to demonstrate the strengths of a middle power country. The same rule applies to the COVAX Facility. Among the East Asian countries, Brunei, China, Japan, Singapore, and the Republic of Korea have signed commitment agreements to the COVAX Facility as self-financing participants. These countries can purchase vaccines when available for up to 20% of their respective populations. Although some of these countries, including Japan, may already have negotiated bilateral purchase agreements with pharmaceutical companies, the additional multilateral COVAX Facility can help hedge the risks related to vaccine procurement. The reason international cooperation is so important is that while some countries like Japan can purchase in advance with a budget, many countries in the same Asian region would have difficulty securing the funding to make such a purchase. Due to the nature of the coronavirus infectious disease, even if a clean area develops in one country, the virus will inevitably be introduced again, so international cooperation and joint response to vaccines and treatments are important.

Middle powers are not superpowers like the United States or China, and although Japan is the third largest economy in the world, it has a peace constitution, which hinders its ability to strengthen its military power. Therefore, the path of human security is not an option but a necessity for middle power countries. These issues comprise limitations to the international cooperation of middle power countries. Of course, there may be logic to talk about the superiority of the logic of power politics in view of the U.S.–China confrontation or the current Ukraine situation, but paradoxically, liberal international cooperation still works, even in the context of Ukraine. The Russia–Ukraine war is leading international relations into a tense phase and could erode the foundation for international cooperation. Under the new context of war, the role of middle power countries to restore international cooperation and maintain peace will grow even more. East Asian countries are relatively less at risk of being involved in international conflicts centred around Russia, given that they are not tied to energy issues with Russia. Middle power countries should pay attention to the risks of involvement in conflict and try to restore the foundation of international

cooperation in areas where cooperation is possible from a humanitarian perspective beyond national interests, such as health security.

Next, following the discussions on the strengths and limitations of middle power countries pertaining to international cooperation as detailed in Sections 3–5, we describe the Asian middle power countries with these strengths and limitations and conclude the paper with a suggestion on the role middle power countries in Asia should play in post-pandemic international cooperation.

## 5. Conclusions

In 2020, humanity suffered a massive infectious disease pandemic that has made history and will long be remembered. When COVID-19 was still being underestimated early in 2020, some argued that the virus would disappear before the summer of that year, while others claimed it was no more serious than the flu and that people were overreacting. However, the COVID-19 pandemic has become the biggest global health and security crisis for all humans since the Spanish flu pandemic in 1918. Whether the end of the pandemic will bring a return to the pre-coronavirus world, even if vaccines and treatments are developed, remains unknown. In the changed world we now live in, a high possibility exists that we will not be able to resume our pre-pandemic life as usual and will have to endure some adjustments. The unknown factor is the direction the world will take in the post-pandemic era. Will nations focus on independent survival? Or will international cooperation be integral to shaping the new world? The world seems to have moved too far forward in globalisation for the national egoism of the super powers to dominate. Even if the virtual replaces the personal and online activities replace offline activities, the fact that humans are social animals remains unchanged, and international cooperation is still valid. In the current context of the COVID-19 pandemic, the two major superpowers, the United States and China, are not performing properly. The United States' response to the pandemic is being criticised as the country continues to record the world's highest number of COVID-19 infections and deaths. At the same time, many countries have developed negative feelings toward China, as China was the geographical starting point of the COVID-19 pandemic, despite the denials of Chinese government officials. Indeed, international organisations, like the two great powers, are not free from criticism for failing to properly fulfil their roles. The European Union, which symbolised free passage as a representative of regional integration, showed only powerlessness through measures such as border closures in the face of COVID-19. The WHO, which is in charge of global health, was not free from criticism either. Therefore, the question remains as to who should promote international cooperation while these powerful countries and international organisations fail to take the necessary leadership role. Due to its inherent disadvantages, it is not free from confrontation between the United States and China, and a conflict of interests exists for each country. Thus, the four middle power countries in Asia, which are economically successful countries, need to play an important role in international society based on their relatively excellent quarantine performance. More specifically, infectious disease crises have been continuously occurring in the realm of health security (SARS, MERS, and COVID-19), and the interval between outbreaks is growing shorter and shorter. What is important in the diplomacy of middle power countries is not traditional security and hard power but the soft power of international law, human rights, health security, and international cooperation. Four Asian countries with the ability to realise these values for the international community must also play a role in the international community after the COVID-19 crisis. The limitations are clear, but the strengths are also clear, so a way to maximise those strengths must be identified, and a path for international cooperation that preserves the strengths and values of middle power countries must be found. The COVID-19 pandemic is an unprecedented disaster in human history, but the very common sense conclusion that the way to overcome it lies in cooperation and mutual trust should not be ignored in international relations. The role of middle power countries in rebuilding international health cooperation is very important, and as seen in the Ukraine crisis, the

eroded foundation of international cooperation can be rebuilt starting with cooperation with middle power countries. This is a niche part that superpowers such as the United States and China do not play, so middle power countries should assume that role.

**Author Contributions:** Conceptualization, Y.K. methodology, Y.P.; software, Y.P.; validation, Y.P.; formal analysis, Y.K.; investigation, Y.K.; resources, Y.K..; data curation, Y.P.; writing—original draft preparation, Y.K.; writing—review and editing, Y.K.; visualization, Y.P.; supervision, Y.K.; project administration, Y.P.; funding acquisition, Y.K. All authors have read and agreed to the published version of the manuscript.

**Funding:** This paper was supported by the KU Research Professor Program of Konkuk University.

**Institutional Review Board Statement:** Not applicable.

**Data Availability Statement:** Not applicable.

**Conflicts of Interest:** The authors declare no conflict of interest.

## Notes

1. See constitution of the World Health Organization (https://www.who.int/health-topics/human-rights#tab=tab_1 (accessed on 27 May 2022)).
2. All data produced by Our World in Data are completely open access under the Creative Commons BY license. You have the permission to use, reproduce, and distribute it in any medium, provided the source and authors are credited.
3. Korea and Japan are among the countries that are in the list of vaccine producing countries with functional national regulatory authorities (NRA). (https://www.who.int/initiatives/who-listed-authority-reg-authorities/list-of-vaccine-prod-countries (accessed on 27 May 2022)).

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
