# Peer review of "International Health Cooperation in the Post-Pandemic Era: Possibilities for and Limitations of Middle Powers in International Cooperation"

_socsci, doi:10.3390/socsci11060259_

Round 1

Reviewer 1 Report

Thank you for the opportunity to review this article, which explores and theorises the possibilities for international health cooperation between nations labelled “middle powers”, drawing on the case studies of Korea, Japan, Singapore and Taiwan in light of the COVID-19 pandemic, examining healthcare diplomacy and security from a multifaceted perspective. This is an interesting paper, both thoughtful and thought provoking. I do find that the piece needs a good deal of further work reinforcing many of the claims throughout with more concrete references to existing literature/studies – the existing reference list is insufficient for a paper or journal of this standard. I also recommend a strengthening of background framing for audiences outside the international relations discipline, given the generalist social science nature of the journal. If these issues can be addressed I would see this as a good addition to the global health politics discourse.

Please see attached for detailed list of suggested revisions.

Author Response

Please look on attached files 

Reviewer 2 Report

Comments  are  in  the attached   file !

Author Response

Please look on attached file

Round 2

Reviewer 1 Report

I thank the authors for their efforts to revise their paper in line with previous comments. The scholarly content of the paper is strong. However there is still a lot of work to do style-wise to bring the paper up to a publishable academic standard.

Much more care needs to be taken to ensure consistency in formatting style for references; select a set style (e.g. APA style) and ensure it is followed thoroughly and consistently. In particular, attention should be paid to multiple citations e.g. p.3 (Funabashi and Ikenberry 2020) (Norrlof 2018) (Drezner 2019) should all be in the same set of parentheses, separated by semi-colons.

Ensure author names are correctly spelled (e.g. p.10 you have "Guinea" but the author's name is "Gunia".) It is not appropriate to cite Wikipedia (p.10) for a scholarly article. 

Where references have been added as sources of figures, these figures need to be captioned/annotated with a citation to THAT source, not another source.

There is still no evidence that you have appropriate permissions to reproduce the graph images in the paper - you must provide this or you cannot use the images.

Round 3

Reviewer 1 Report

Thank you very much to the authors for engaging in the improvements suggested last time. The standard of the paper has been greatly improved and I am happy to recommend proceeding to acceptance.